# Evaluation of Sperm Mitochondrial Function: A Key Organelle for Sperm Motility

**DOI:** 10.3390/jcm9020363

**Published:** 2020-01-29

**Authors:** Federica Barbagallo, Sandro La Vignera, Rossella Cannarella, Antonio Aversa, Aldo E. Calogero, Rosita A. Condorelli

**Affiliations:** 1Department of Clinical and Experimental Medicine, Policlinico “G. Rodolico”, University of Catania, 95123 Catania, Italy; federica.barbagallo11@gmail.com (F.B.); rossella.cannarella@phd.unict.it (R.C.); acaloger@unict.it (A.E.C.); rosita.condorelli@unict.it (R.A.C.); 2Department of Experimental and Clinical Medicine, “Magna Graecia” University, 88100 Catanzaro, Italy; aversa@unicz.it

**Keywords:** sperm motility, asthenozoospermia, mitochondrial function, antioxidants, prokinetic therapy, male infertility

## Abstract

Introduction: The role of nutraceuticals in the treatment of male infertility, especially in the “idiopathic form”, remains the subject of significant debate. Many antioxidants improve sperm motility but the exact mechanism by which they act is still unclear. Although several studies have shown a correlation between sperm motility and mitochondrial function, the effects of antioxidant therapy on mitochondrial membrane potential (MMP) are poorly studied. The first aim of this review was to evaluate the efficacy of antioxidants on mitochondrial function and, consequently, on sperm motility in male infertile patients. Material and Methods: we performed a systematic search of all randomized controlled and uncontrolled studies available in the literature that reported sperm motility and MMP at baseline and after antioxidant administration in-vivo and in-vitro in patients with idiopathic asthenozoospermia. Pubmed, MEDLINE, Cochrane, Academic One Files, Google Scholar and Scopus databases were used. Results: Unexpectedly, among 353 articles retrieved, only one study met our inclusion criteria and showed a significant effect of myoinositol on both MMP and sperm motility. We then summarized the main knowledge on anatomy and metabolism of sperm mitochondria, techniques allowing to assess sperm mitochondria function and its relationships with low sperm motility. Finally, we paid special attention to the effect of antioxidant/prokinetic molecules for the treatment of asthenozoospermia. Conclusions: This is the first systematic review that has attempted to evaluate the effects of antioxidants on MMP and sperm motility. Although results are not conclusive due to the dearth of studies, the close relationship between mitochondria and sperm motility is clear. The investigation of this correlation could provide valuable information to be exploited in clinical practice for the treatment of male infertility.

## 1. Introduction

Sperm motility is a fundamental requirement to ensure male fertility. Studies and interest for sperm motility started in 1919 when Lillie Frank Rattray, an American zoologist, author of the book “Problems of fertilization”, for the first time, talked on the energetic metabolism of spermatozoa. He said: “Spermatozoa are probably incapable of receiving nourishment outside of the gonad after they are fully differentiated; certainly in the case of external insemination there is no opportunity for the restitution of substance …”. Since those years, several studies focused on the “power plant” of the cell, the mitochondrion, demonstrating the key role of this organelle on cellular homeostasis and sperm motility [1]. On this account, the mitochondrion role in male fertility cannot be ignored: it is fundamental not only for sperm motility but also for hyperactivation, capacitation, acrosome reaction and fertilization. Thus, recent research in sperm physiology is focusing on this important organelle, as a biomarker of sperm health and fertility [2,3,4,5]. Mitochondrial sperm dysfunction is also implicated in the pathogenesis of seminal oxidative stress, a key element responsible of many cases of “apparently” idiopathic male infertility [6].

In recent decades, the role of nutraceuticals in the treatment of male infertility has become the subject of significant debate. Many antioxidants are used in clinical practice to improve sperm parameters. Although several studies have shown a correlation between sperm motility and mitochondrial function, the effects of antioxidant therapy on the mitochondrial membrane (MMP) have rarely been studied. The first aim of this review was to evaluate the efficacy of antioxidants on mitochondrial function and, consequently, on sperm motility in male infertile patients.

## 2. Methods

### 2.1. Sources

Data were independently extracted by F.B. and R.A.C. A systematic search was performed through Pubmed, MEDLINE, Cochrane and Scopus databases, from each database inception to 30 Nov. 2019, using Medical Subjects Headings (MeSH) indexes and key words searches. 

The search strategy used combined MeSH terms and key words and was based on the following key words: sperm motility, asthenozoospermia, mitochondrial function, mitochondrial activity, mitochondrial membrane potential, antioxidants, nutraceuticals, male infertility, prokinetic therapy, carnitine, myoinositol, coenzyme Q10. Additional manual searches were made using the reference lists of relevant studies. 

### 2.2. Study Selection

Studies that met the following inclusion criteria were included in the qualitative synthesis to focus on the effect of antioxidant therapy with prokinetic effect on mitochondrial function and consequent on sperm motility: Design: randomized controlled and not-controlled clinical trial;Patients inclusion criteria: idiopathic asthenozoospermiaPatients exclusion criteria: known causes of male infertility such as male accessory gland infection, varicocele, hypogonadism, Y chromosome microdeletions, testicular torsion or trauma, history of cryptorchidism, as well as thyroid, pituitary or adrenal disorders, liver or kidney failure.Study intervention: antioxidants administration in vivo or in vitroStudy outcome: both sperm motility and mitochondrial membrane potential evaluated after antioxidant administration, compared to those of a treated or not-treated control group or with baseline. Spontaneous pregnancy.

## 3. Results

Using the aforementioned search strategy, 353 articles were retrieved. After excluding duplicate records, 283 articles were screened. Of these, 239 were found to be irrelevant after reading their abstract. The remaining 44 full-texts were carefully read. Among these, 38 studies were excluded because only sperm motility was assessed and not mitochondrial function; four studies were excluded because no exclusion criteria were used for patient selection and one study did not report results on sperm motility. Finally, only one study met our inclusion criteria for a total of 20 patients (Figure 1). This is the first systematic review evaluating the effects of antioxidant therapy on MMP and sperm motility assessed in the same study. However, the results cannot be conclusive due to the lack of an adequate number of studies and patients.

Before discussing the results of the effects of prokinetic therapy on mitochondrial function and sperm motility, in Section 4, Section 5, Section 6 and Section 7 we have summarized the main knowledge related to sperm mitochondria and its correlation with sperm motility for provide the basis for future studies that are absolutely necessary. Investigation of this correlation could provide valuable information to be exploited in clinical practice for the treatment of male infertility.

## 4. Sperm Mitochondria: Anatomy 

Mammalian spermatozoa typically have between 50 and 75 mitochondria [7]. Mitochondria of spermatozoa show peculiar characteristics. They are exclusively confined in the mid-piece, tightly wrapped around the axoneme. During spermiogenesis, the mitochondria line up end-to-end and wrap helically around the flagellum to form the thick mitochondrial sheath, just under the outer plasma membrane of the cell [8,9]. The mitochondrial capsule is thus shaped by multiple disulphide bridges formed between cysteine and proline-rich selenoproteins [10,11,12]. This intricate attachment to the fibrous sheath makes sperm mitochondria particularly difficult to isolate by conventional separation method [13]. Moreover, many proteins and enzymes such as subunit VIb of the cytochrome oxidase [14], E1-pyruvate decarboxylase and creatine kinase (CK) [15,16] are isoforms present only in sperm mitochondria. 

## 5. Sperm Mitochondrial Metabolism 

### 5.1. A Long Debate: Glycolysis or Oxidative Phosphorylation? 

Many studies on sperm mitochondrial bioenergetics have concentrated on the following question: what is the main biochemical pathway that provides the energy for sperm motility, glycolysis or oxidative phosphorylation (OXPHOS)? Studies carried out in several species, including humans, have often provided different and/or conflicting results. Many authors have reached the conclusion that glycolysis has a primary role in energy production in human sperm motility [17,18,19,20]. Other authors have underlined the importance of mitochondrial OXPHOS for sperm motility [21,22,23,24,25,26,27,28]. Moreover, the experimental conditions varied significantly from one study to another and this complicated the interpretation of the results. Therefore, despite numerous studies, a clear conclusion cannot be drawn [12]. A reasonable concept that emerged from many studies is that these processes are not mutually exclusive and that spermatozoa exhibit a great versatility in their metabolism using glycolysis exclusively, mitochondrial OXPHOS exclusively or a combination of both pathways for energy production according to the substrates available in the female genital tracts [29,30]. Specifically, Zhu et al. had recently shown that mitochondrial oxidative phosphorylation is activated to produce ATP under low glucose condition. They incubated boar spermatozoa with different doses of glucose and they found that sperm progressive motility and straight-line velocity were significantly increased with decreasing glucose level in the incubation medium. They also showed that, in presence of the mitochondrial translation inhibitor d-chloramphenicol, mitochondrial protein synthesis, mitochondrial activity and ATP level were suppressed, and consequently, the linear motility speed decreased. Interestingly, despite the reduction of linear motility speed, total motility did not change [31]. These results suggest that sperm motility patterns depend on the substrates available and as a result on the biochemical pathway that is activated. While glycolysis is important for hyperactivated motility [32], the high-speed linear motility is induced via activating the mitochondrial activity in low glucose condition.

### 5.2. Reactive Oxygen Species and Sperm Mitochondria

Since mitochondria have a main role in sperm metabolism and energy production, they are the major reactive oxygen species (ROS) generator, as they convert approximately 1–2% of consumed oxygen into superoxide anions [33]. In spermatozoa, mitochondrial Complex I and Complex III are the major sites for ROS production [33]. An imbalance between impaired ROS production and antioxidants mechanisms may be extremely harmful for spermatozoa [34]. These cells are particularly susceptible to oxidative stress because they cannot restore damage caused by oxidative stress due to deficiency of cytoplasmic repair enzymes [35]. In contrast, at physiological concentrations, ROS are trigger for several reproductive mechanisms such as sperm capacitation, hyperactivation, acrosome reaction and oocyte fusion [36,37,38]. More recent research is changing the long-accepted dogma that ROS is ultimately a negative indicator of sperm function, thus indicating that ROS production can also reflect intense mitochondrial activity leading to increased sperm function [5]. 

## 6. Techniques to Study Sperm Mitochondria Function

Since sperm mitochondria have been related to sperm motility and fertilization, numerous tools have been developed to evaluate their function. Several parameters can be used to study sperm mitochondria. This include mitochondrial activity, MMP levels and mitochondrial calcium levels [39]. The use of fluorescent probes to detect changes in the MMP is the most popular tool used to evaluate mitochondrial function. These probes spread freely through the plasma membrane to the cell cytosol and accumulate electrophoretically in the mitochondrial matrix, due to the motive force of the proton and acting according to the ability of the mitochondria to pump the protons from the matrix to the intermembrane area [40,41,42]. Therefore, according to their properties, high concentrations of these fluorochromes accumulate in hyperpolarized mitochondria (high MMP), while lower concentrations are found in depolarized mitochondria (low MMP) and the intensity of the fluorescence correlates with MMP [43].

Many commercial fluorescent dyes are widely used, such as JC-1 (5,5,6,6-tetrachloro-1,1,3,3-tetraethylbenzimidazolyl carbocyanine iodide [41], Mito Tracker Green FM [44] and rhodamine 123 [45]. The JC-1 probe is one of the preferred fluorescent dyes for analyzing MMP and has been widely used for the analysis of spermatozoa in several species including humans [28,46], cattle [47], horse [48], ram [49], dog [50] and alpaca [51]. In 2004, Marchetti et al. compared the specificity of four fluorochromes in the evaluation of sperm MMP. They found that JC-1 is found only within the mitochondria and therefore provides the most accurate measurement of MMP [26]. Lugli et al. confirmed this finding, indicating that JC-1 is more reliable and specific for this type of evaluation than other probes [52]. Amaral and Ramalho-Santos found JC-1 more dynamic and suitable than MitoTracker Green and MitoTracker Red, as it is able to detect minimal changes in MMP [11,53]. Recently, Uribe et al. evaluated the usefulness of another fluorescent dye, tetramethyl rhodamine methyl ester perchlorate (TMRM) for measuring sperm MMP. They found that TMRM is able to accurately detect MMP variations comparable to the method widely used, JC-1 staining. In addition, TMRM was able to measure sperm MMP in the experimental conditions in which JC-1 had previously presented difficulties [54].

Finally, mitochondrial oxygen consumption is considered the central parameter of mitochondrial function. Studies on experimental animals have shown that mitochondrial oxygen consumption is positively correlated with traditional measures of sperm function including motility and vitality [2]. Therefore, measurement of oxygen consumption is another important biological endpoint for the study of sperm mitochondrial function and should be further investigated in future studies.

## 7. Asthenozoospermia and Sperm Mitochondrial Dysfunction 

Almost forty years have passed since the first study relating mitochondrial function to sperm motility. Everson et al. reported a good correlation between sperm motility and MMP, comparing ejaculate from fertile men with those from patients whose spermatozoa showed reduced sperm motility [55]. Ruiz-Pesini et al. compared mitochondrial respiratory complex activities of 86 asthenozoospermic patients with those of 33 controls. Their results showed that semen samples of controls had substantially higher activities of complexes I, II and IV compared with those of patients with asthenozoospermia. Moreover, a direct and positive correlation was found in the whole population studied between spermatozoa motility and all the mitochondrial respiratory complex activities assayed (I, II, I+III, II+III, and IV). They suggested that more specific mitochondrial dysfunctions could be the underlying cause of idiopathic asthenozoospermia [24]. In a following study, these authors found that mitochondrial enzyme activities not only correlate with sperm motility but also with vitality and cell concentration [25]. As mitochondria are the major source of pro-oxidative agents, it is suggested that dysfunction of this organelle would have a fundamental role in the oxidative imbalance affecting sperm function [33]. Wang et al. [56] identified low MMP and high ROS production in spermatozoa from infertile patients, probably as a consequence of such mitochondrial injury. Other researchers have observed changes in mitochondrial function in sperm derived from infertile patients [57]. However, spermatozoa with high MMP have been identified in fertile men [26,27,58,59]. Paoli et al. [46], correlated MMP with increasing motility to establish MMP values corresponding to a precise sequence of gradually increasing motility. They evaluated 185 semen samples, divided into 13 motility classes (from 0 to 60%) with an increment of 5% between classes and a second group of 28 semen samples showing nonlinear motility only, divided into five classes. A positive correlation was found between sperm motility and FL2 (percentage of sperm with high and low MMP) in all samples of both groups. In group A, the first 0 motility class, showed a mean FL2 of 11.5. This value increased gradually with increasing motility, reaching its highest mean value of 75.7% in the 60% motility class. They found both immotility and severe asthenozoospermia to be characterized by an extremely low MMP. These results diverge from those of Piasecka, whose study on 32 subjects with normal motility and 25 patients with asthenozoospermia identified two asthenozoospermic subpopulations, one with a low and the other with a high MMP, suggesting that the reduced motility might be caused not only by functional alteration of mitochondria but also by abnormal morphology of the axoneme, the dense fibers and the fibrous sheath [60]. Pelliccione et al. demonstrated the importance of structural defects in the mitochondrial membrane in asthenozoospermic semen samples, indicating a close correlation between forward motility and the percentage of the intermediate tract having a normal membrane [61].

Over the last years, proteomic studies have tried to identify dysfunctional proteins responsible for asthenozoospermia [62,63]. Recently, Nowicka-Bauer et al. compared proteomic profiles in spermatozoa of normozoospermic men and in patients with isolated asthenozoospermia. The results of this study were further supported by two additional mitochondrial tests (JC-1 and MitoSox Red) to establish a possible direct connection between the identified proteins and functional status of mitochondria [63]. In accordance with a previous study by Amaral et al. [62], they found that most of the identified dysfunctional proteins in low motility spermatozoa were of mitochondrial origin and a great proportion of them were engaged in cell metabolism and energy production being involved in tricarboxylic acid cycle (TCA), mitochondrial OXPHOS and the metabolism of butanoates, propanoates and pyruvates. These results strongly support the emerging idea that several bioenergetics metabolic pathways contribute to sperm motility (de)regulation [62,63]. Table 1 summarizes the main studies that investigated the correlation between human sperm mitochondrial function and motility.

## 8. Effects of Prokinetic/Antioxidant Therapy on Sperm Mitochondrial Function and Motility

The first aim of this review was to evaluate the efficacy of prokinetic/antioxidant administration on mitochondrial function and sperm motility in infertile patients. There are several antioxidants commonly used to improve male fertility. Among them, the most used are zinc, folic acid, N-acetylcisteine, coenzyme Q10, vitamins E and C, selenium, carnitines and pentoxifylline in various dosage and variable combinations. However, the most recent Cochrane systematic review noted that many studies currently available are of poor quality [64]. A position statement by the Italian Society of Andrology and Sexual Medicine gave a high value when antioxidants were administered in patients with idiopathic infertility in the presence of documented abnormal sperm parameters and alteration of sperm DNA fragmentation after adequate diagnostic procedure [65]. A further limitation is the almost total absence of comparative studies between the various antioxidants and/or their combinations. Therefore, it is not possible to suggest specific antioxidants, detailed treatment regimens or to indicate rates in terms of improvement of sperm parameters and pregnancy rate [65]. Many antioxidants improve sperm motility but the exact mechanism by which they act is still unclear. As previously described, ROS are normally generated during mitochondrial oxidative phosphorylation (OXPHOS) but high levels of these compounds may impair mitochondrial activity, and consequently sperm motility. Zhu et al. have recently reported that ROS impairs both ATP production and mitochondrial transcription, and consequently, they have a negative effect on linear motility in boar spermatozoa. Interestingly, they also showed that treatment with PQQ and CoQ10 increased mitochondrial DNA (mt-DNA) stability and countered the damage caused by ROS to the specific RNA polymerase (POLMRT) and the mitochondrial transcription factor-A (TFAM) that are fundamental for mitochondrial transcription. These results suggested that a mitochondrion-targeted antioxidant treatment could improve sperm linear motility by preserving the sperm transcription system from ROS damage [66].

Despite this close correlation between mitochondrial function and sperm motility, there are no randomized clinical trials in patients with idiopathic asthenozoospermia, which assessed both sperm parameters and MMP after administration of antioxidants. Only few in-vitro studies have evaluated both MMP and sperm motility in patients with OAT after incubation with antioxidants [59,67,68,69,70,71]. Raigani et al. conducted a double-blind, randomized, placebo-controlled study of 83 subfertile OAT patients to evaluate the effects of oral supplementation of folic acid and zinc sulphate on both sperm motility and MMP. The results of this study showed that zinc sulphate and folic acid supplementation did not improve sperm quality in infertile patients with severely compromised sperm parameters [69]. Zhang et al. [59] studied the effects of l-carnitine (LC) to prevent sperm damage during the freeze-thaw process in 37 astheno- and 33 normo-zoospermic human semen samples. They assessed motility, viability, MMP and DNA fragmentation index (DFI) on fresh sperm aliquots, frozen-thawed control and samples treated with LC. They found that the integration of the cryopreservation medium with LC improved significantly fast forward motility, forward motility, total motility and viability in post-thawing spermatozoa in both asthenozoospermic and normozoospermic semen samples compared to controls.

Ghafarizadeh et al. analyzed semen samples from 50 OAT patients after incubation with 2 μg/mL of selenium at 37 °C for 2, 4 and 6 h. They demonstrated that total and progressive sperm motility, sperm viability and MMP were significantly higher in selenium-treated samples than in the control group after a 4- and 6-h incubation. In this study, selenium supplementation in-vitro protects spermatozoa from mitochondrial damage, probably due to the antioxidant properties of selenium, leading to a reduction in ROS [70]. However, the studies mentioned above did not use the exclusion criteria for patient selection. Therefore, only one study met our inclusion criteria [67]. Spermatozoa from 20 normozoospermic men and 20 patients with OAT were incubated in-vitro with 2 mg/mL of myoinositol or phosphate-buffered saline as a control for 2 h. Myoinositol is a component of the vitamin B complex and it is the most biologically important form of inositols in nature. They found that spermatozoa from patients with OAT incubated with myoinositol had a significantly higher progressive and total motility and a higher MMP. Moreover, in patients with OAT, the percentage of spermatozoa with low MMP was significantly lower (*p* < 0.05). The effect of myoinositol, probably increasing cytosolic Ca++, and consequently, the inner mitochondrial Ca++, seems to act specifically on mitochondrial function, because others bio-functional sperm parameters did not show any significant variation. Other in vitro [72,73,74] and in vivo [75,76,77,78] studies have shown that myoinositol, alone or in combination, increased sperm motility, but the study published by Condorelli et al. in 2012 is the only one that had simultaneously evaluated sperm motility and MMP.

## 9. Conclusions

The close relationship between mitochondria and sperm motility is clear but unfortunately not yet effectively exploited in clinical practice. Many dark points have yet to be clarified. First of all: who are the patients who would benefit from the MMP assessment? When and which bio-functional sperm parameters are useful to be evaluated? Furthermore, an important limitation is the lack of randomized clinical trials for the evaluation of mitochondrial function in patients with idiopathic asthenozoospermia after antioxidant/prokinetic therapy. Future studies are needed to evaluate the effects of antioxidants on MMP and, consequently, on sperm motility, and to understand the mechanisms by which they act. MMP assessment could complement standard semen analysis and help identify adequate and personalized treatment for each patient, particularly in the form of apparently idiopathic male infertility.

## Figures and Tables

**Figure 1 jcm-09-00363-f001:**
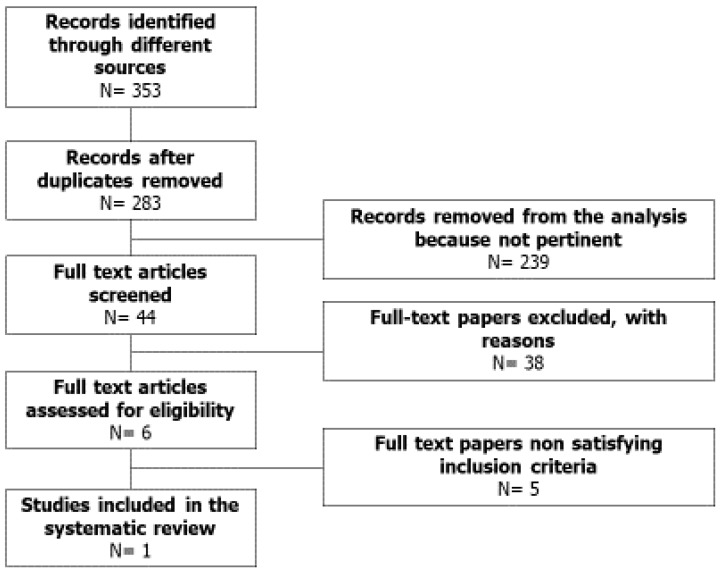
Flowchart of the studies included in systematic review.

**Table 1 jcm-09-00363-t001:** Studies on mitochondrial dysfunction and sperm motility.

Authors	Method	Results	
		Mitochondria	Sperm Motility
Everson et al., 1982 [55]	Determination of MMP with Rhodamine 123	Reduced MMP	Asthenozoospermia
Ruiz-Pesini et al., 1998 [24]	Determination of enzymatic activity by spectrophotometric assay	Reduced activity of Complexes I, II and IV and of citrate synthase	Asthenozoospermia
Ruiz-Pesini et al., 2000 [25]	Determination of enzymatic activity by spectrophotometric assay	Reduced activity of Complexes I, II, I+III, II+IV and IV and of citrate synthase	Asthenozoospermia
Troiano et al., 1998 [57]	Determination of MMP with JC-1	Reduced MMP	Asthenozoospermia
Marchetti et al., 2002 [26]	Determination of MMP	High MMP and low DNA fragmentation	High motility
Kasai et al., 2002 [58]	Determination of MMP with JC-1	High MMP	High motility
Wang et al., 2003 [56]	Determination of MMP with DiOC6(3) and ROS with chemiluminescence assay using luminol.	Reduced MMP Higher ROS levels	Asthenozoospermia
Piasecka et al., 2003 [60]	Determination of MMP with JC-1 and Mito Tracker Green FM	Two subpopulations, one with a low and the other with a high MMP	Asthenozoospermia
Gallon et al., 2006 [27]	Determination of MMP with DiOC6(3)	High MMP	High motility
Ferramosca et al., 2008 [13]	Determination of oxygen consumption by polarographic assay	Reduction of respiratory control ratio	Asthenozoospermia
Paoli et al., 2011 [46]	Determination of MMP with JC-1	Increasing MMP Low MMP	13 motility classes (from 0–60%) with an increment of 5% between classes Immotility or severe asthenozoospermia
Pelliccione et al., 2011 [61]	Analysis of tail middle piece (MP) by transmission electron microscopy	Structural defects in mitochondrial membranes	Asthenozoospermia
Zhang et al., 2016 [59]	Determination of MMP with JC-1; mitochondrial DNA copy number (mtDNAcn), mtDNA integrity analysis using long PCR and apoptotic parameters	High MMP Low mtDNAcn High mtDNA integrity	High motility
Amaral et al., 2014 [62]	Proteomic study	Dysregulation of mitochondrial proteins	Asthenozoospermia
Nowicka-Bauer et al., 2018 [63]	Proteomic data supported by two additional mitochondrial tests (JC-1 and MitoSox Red)	Dysregulation of mitochondrial proteins	Isolated Asthenozoospermia

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
