# Peer review of "Evaluation of Sperm Mitochondrial Function: A Key Organelle for Sperm Motility"

_jcm, 2020, doi:10.3390/jcm9020363_

Round 1

Reviewer 1 Report

The review deals with a very important aspect of male infertility, especially the idiopathic one, whose cause should be sought at the cellular and molecular level. In many cases determination of etiopathogenesis and treatment of asthenozoospermia is difficult and requires proven research methods. The authors focused on assessing the sperm mitochondrial function, because abnormal function of these organelles leads to low sperm motility. The authors conducted a systematic search of studies revealing the correlations between sperm motility, mitochondrial membrane potential (MMP), antioxidant therapy and idiopathic asthenozoospermia. Pubmed, MEDLINE, Cochrane, Academic One Files, Goole Scholar and Scopus databases were used. The methods and results have been described correctly. On the basis of the data obtained, the authors summarized the knowledge concerning anatomy and metabolism of sperm mitochondria, techniques allowing to assess the function of sperm mitochondria and relationships between low sperm motility and function of sperm mitochondria. In addition, the authors paid special attention to the effect of antioxidant therapy/prokinetic in the treatment of asthenozoospermia. The authors were critical of the results obtained and raise very important issues in the Conclusion chapter. One of them is related to the clinical application of studies on MMP assessment and antioxidant therapy.

In summary, the manuscript highlights the important role of assessing the function of sperm mitochondria in establishing and treating asthenozoospermia.

Reviewer remarks:

I suggest that the abstract should be rewritten in accordance with the chronology of the data contained in the manuscript: background, methods, results, conclusions; The goal of the study is not described in the abstract (the aim is provided only at the end of the Introduction); Table 1 and Figure 1 are not cited in the text; “sperm” should be added in title of subunit 7 and 8: “… sperm mitochondrial function, “…on sperm mitochondrial function and motility”

Reviewer 2 Report

This review focus on the mitochondrial function of sperm, especially its roles in sperm motility. The authors introduced the activation of mitochondria in motile sperm, ROS generation, the dyfunction of sperm mitochondria in infertile man and the effects of antioxidants on sperm motility. They picked up articles about human sperm study and then discussed sperm mitochondria functions described above. The information is valuable and will be contributed for understanding the mechanisms of sperm mobility and the relationship between low quality of sperm and dyfunction of sperm mitochondria. However, there is few information to clear the mechanisms in more detail. The reviewer thinks that the authors should mention not only human sperm but also other animals. For example, Zhu et al. reported using boar sperm that the dose of glucose in media regulates glycolytic pathway and mitochondria in sperm (Zhu et al., Frontia Physiology). The different metabolic pathway changes sperm motility pattern. They also clearly showed the targets of oxidative stress in sperm mitochondria (Zhu et al., Free Radical Biology and Medicine). The information support to understand the mitochondria functions in human sperm.

The authors should introduce human sperm information and the discuss about them using articles of other animals sperm and other cell types.

Round 2

Reviewer 2 Report

According to reviewer's comments, the authors revised their manuscript well.